# Effects of Recycled Rubber Particles Incorporated as Partial Sand Replacement on Fresh and Hardened Properties of Cement-Based Concrete: Mechanical, Microstructural and Life Cycle Analyses

**DOI:** 10.3390/ma16010063

**Published:** 2022-12-21

**Authors:** Natividad Garcia-Troncoso, Samantha Acosta-Calderon, Jorge Flores-Rada, Haci Baykara, Mauricio H. Cornejo, Ariel Riofrio, Katherine Vargas-Moreno

**Affiliations:** 1Facultad de Ingeniería en Ciencias de la Tierra (FICT), Escuela Superior Politécnica del Litoral, ESPOL, Campus Gustavo Galindo Km 30.5 Vía Perimetral, Guayaquil P.O. Box 09-01-5863, Ecuador; 2Centro de Innovación Holcim, Holcim Ecuador S.A., Guayaquil 090616, Ecuador; 3Facultad de Ingeniería Mecánica y Ciencias de la Producción, Escuela Superior Politécnica del Litoral, ESPOL, Campus Gustavo Galindo Km 30.5 Vía Perimetral, Guayaquil 090112, Ecuador; 4Center of Nanotechnology Research and Development (CIDNA), Escuela Superior Politécnica del Litoral, ESPOL, Campus Gustavo Galindo Km 30.5 Vía Perimetral, Guayaquil 090506, Ecuador; 5Escuela de Ingeniería Civil, Universidad Espíritu Santo (UEES), Samborondón P.O. Box 09-01-952, Ecuador

**Keywords:** rubber concrete, life cycle assessment, mechanical properties, rubber particles

## Abstract

Cement is one of the most valuable materials in today’s society, as it is used in most construction developments known to mankind. However, the energy intensive process and significant environmental impacts related to the production of Ordinary Portland Cement have shown the importance of searching for more sustainable materials. Concrete uses different aggregates added to the cement binder to lower, not only cost, but other factors like environmental burden, while maintaining good mechanical properties. This study analyzes the properties of fresh and hardened concrete incorporating recycled rubber to replace fine aggregate. Locally sourced 2 mm diameter rubber was incorporated in a regular strength concrete matrix into three different replacement levels, i.e., 3%, 5%, and 10%. Compression, tensile, flexural, and modulus of elasticity of hardened concrete were carried out in specimens aged 7, 14, and 28 days. In addition, non-destructive ultrasonic pulse velocity and rebound number tests were only performed on specimens aged 28 days. Once the tests were carried out, the fresh and hardened concrete properties were obtained. Similarly, the compressive and flexural strengths had the exact relationship between the values obtained. On the other hand, the modulus of elasticity tends to decrease due to the presence of the rubber. Consequently, it is recommended not to develop mix designs with more than 5% rubber because it is not meaningfully affected. The fine aggregate can be partially replaced by the rubber, keeping almost the same performance compared with sand-only counterparts. In addition, the life cycle assessment showed a reduction of up to 40% in the global warming potential. In fact, the 15% recycled rubber concrete mix has a climate change indicator of approximately 245 kg of CO_2_ eq.

## 1. Introduction

Over the years, the construction sector has been a critical factor for economic and social development in a determinate place [1], where concrete has been one of the most demanded materials for construction [2]. The components that make up concrete are water, cement, aggregates, and additives; however, cement and aggregates are the most important contributors to generate a more significant environmental impact throughout their manufacture [3,4].

The raw materials for cement as well as aggregates that come from natural resources are generally over-exploited [5,6,7,8]. Specifically, Portland cement is associated with high carbon dioxide emissions, responsible for 3.8% of the net global greenhouse gas and representing ca. 5% of global carbon dioxide emissions [9,10,11,12,13]. On the other hand, aggregate quarrying is directly related to opencast mining [14], leading to environmental impacts such as soil fertility and air pollution [15,16], and finally affecting the local ecosystem.

Due to the meaningful environmental impact generated by the aggregates, several studies have been carried out to propose potential replacements for coarse and fine aggregates by using more environmentally friendly substitutes. For coarse aggregates, these can be replaced with recycled concrete [17], coconut shell [18], porcelain, red ceramic [19], palm kernel shell [20], bricks [21], oyster shell [22], seashell [23] and many other materials. In the case of fine aggregate, volcanic ash [24], polyethylene terephthalate particles [25], cooper slag [26], plastic [27], cockle shell [28], rubber [29], and others have been proposed.

Several studies have shown that a substitute for the aggregate may not meaningfully alter the mechanical properties of the concrete, depending on the type of substitute and quantity such that the concrete may keep the same mechanical characteristics or vary slightly [30,31,32]. Regarding the environmental impact generated by concrete partially replaced by by-products, replacing a naturally occurring aggregate reduces the exploitation of the mineral resources, and consequently, the carbon emissions are reduced as well [33]. Regarding the economic issue, the cost of concrete is reduced because of replacement; however, it depends strongly on the cost of processing to meet technical requirements in the concrete industry. On the other hand, Idris et al. [34] and Asutkar et al. [35] reported that incorporating rubber instead of fine aggregate in concrete could be a possible solution to reduce plastic pollution and obtain a block of concrete with good mechanical properties.

This research discusses the mechanical properties of fresh and hardened concrete, including resistance to compression, tension, and ultrasonic pulse velocity. The outcomes of this research work provide meaningful information concerning the fundamental properties of the concrete material using recycled rubber as a replacement for the fine aggregate. The performance of the fresh and hardened concrete was determined through the slump, air content, pressure, and destructive and non-destructive tests. In addition, the environmental impact of the 1 m^3^ recycled rubber-reinforced concrete production has also been evaluated using life cycle analysis. Considering better mechanical properties obtained, recycled rubber can be used in concrete reinforcement, and it can reduce the environmental impact by taking the life cycle point of view into account.

## 2. Materials and Methods

### 2.1. Materials

The study reveals the use of recycled rubber in the concrete mixture, which acts as a substitute for part of the fine aggregate according to the percentage established in each mixture. This material is vulcanized rubber granules and powder from mechanical temperature crushing environmental, end-of-life tires. For recycled rubber concretes to test, Holcim’s hydraulic cement was used in this study. The cement is commercially known as HE cements, which refers to “high early-strength”, and provides more resistant mixes to all ages than conventional general-purpose cement. According to the datasheet provided by Holcim [33], this material was manufactured under the Ecuadorian standard NTE INEN 2380, which is equivalent to the American standard ASTM C1157 [36]. This standard indicates specifications for general and special applications of the cement HE (Table 1).

The aggregates used in the mix were of three types: one type of coarse aggregate and two types of fine aggregate. The coarse aggregate used was No.67 according to ASTM C-33 [37], which indicates the maximum and minimum size of the aggregates as 25.4 mm (3/4”) and 2.36 mm (#8 mesh), respectively. The fine aggregates were a river and crushed sand, with similar grain sizes ranging between 4.75 mm and 0.0015 mm. Each fine aggregate meets the technical requirements for sand in concrete. In fact, the river sand provides compactness and the other the required resistance.

Additionally, two chemical additives were used, a retarder and a superplasticizer. The retarder, commercially known as Sikaplast 2001-R, is a water reducer retardant fabricated to meet the technical requirement of standard ASTM C-494 type D [38]. It mainly consists of polycarboxylates and has a density of approximately 1.05 kg/L. According to the datasheet from Sika [39], it is suggested to be used at an optimum level of 0.5%–0.8% by weight of cement. On the other hand, a superplasticizer, Sikaplast–5012, was used for better workability according to ASTM C–494 type A and F [40]. It has a density approximately of 1.1 kg/L. The datasheet from Sika [34] suggests using a dose of approximately 1–1.5% of the weight of the cement.

The procedure to obtain recycled rubber particles is as follows: first, end-of-life tires are received, then classified according to the type from which they came (light car or heavy vehicle). Afterward, two types of shredding are carried out, one primary and one secondary [41]. The primary crushing is carried out in a machine where pieces of approximately 300 mm are obtained. On the other hand, secondary crushing is done on a mechanical grill to calibrate the size of the output material and obtain the pieces with an average size of 300–500 mm. The next step is demetallization, in which the metal is separated from the material, then it is the process of granulators (primary and secondary) which reduce the chunks that come from the secondary crusher to an average size of 0 to 4 mm. Once the desired size is obtained, the next step is to the fiber aspirator, where the nylon fiber is separated from the rubber grains. Finally, the material passes through some particle separation meshes to be classified and the larger particles go back through the secondary crushing. According to the datasheet from ECSADE S.A. [42], the material is black, has a solid form of granules, and has a characteristic rubbery smell (see Figure 1). The density is in the range of 0.7942 and 1.032 g/cm^3^, the specific weight is 1.15 to 1.27, and it has a percentage of humidity less than 0.75. According to the datasheet from ECSADE S.A. [42] the rubber particle size gradation is from 0–3 mm and the commercial name is Rubber-FLEX.

### 2.2. Methods

For aggregates to characterize, tests were run to measure properties such as fineness modulus, specific weight, moisture content, and absorption percentage. In the case of the moisture content, it is a value needed only from coarse aggregate; therefore, it is not performed on fine aggregate. The value was used to adjust materials quantities in the mix. The test procedure is as follows: first, determine the sample size. According to ASTM C566 [43] the sample size depends directly on the nominal maximum size of the aggregate. Hence, the coarse aggregate No.67 is 19 mm. Once the sample is weighed it goes to an oven, commonly at 110 ± 5 °C for 24 h. After the required time in the oven has elapsed the sample is reweighed, and finally the calculations are done according to the standard method. The sample’s weight before the oven was 3000.5 g, and after the oven was 2962.4 g; therefore, the moisture content was 1.27%.

Like the moisture test, the absorption test is used to adjust material quantities in the mixture, especially in amounts of water. Since this test is used to calculate the percentage of water, the aggregate can absorb it while it is in contact. Following the recommendations of the ASTM C127 [44], the procedure can only be performed in the coarse aggregate. Additionally, as the sample is related to the nominal maximum size the sample is the same as in the previous test. The procedure is to immerse the sample in water for 24 ± 4 h, remove it from the water, dry it superficially and weigh it. The next step is to compare and weigh a sample that has been in the oven for 24 h (it can be the same sample from the previous moisture content test). Finally, the respective calculations from the standard are used to determine the water absorption percentage. The final value calculated was 1.30% of absorption for the coarse aggregate.

The nominal size of an aggregate is necessary for many aspects. In the same way, it is essential to know the distribution of particle sizes within a sample. For this study, the granulometry test was carried out for coarse and fine aggregates to identify the distribution and the fineness module (Figure 2). The standard used for this test was the ASTM C136 [45], which describes taking a sample (previously dried in the oven) and placing it in a mechanical sieve shaker. It is essential to mention that there is a mechanical sieve shaker according to the type of aggregate (coarse or fine) in which the diameter of the sieve is different. This machine is used to drive the movement of the particles from side to side so that the material moves in different orientations to pass to the next sieve. Once the mechanical sieve stops, the next step is to weigh the stock material in each sieve and perform the granulometric curve.

Once the aggregates were tested mix design proceeded using the ACI method [46], in which the design strength is 21 MPa. In this study, four designs were considered: the first is the nominal concrete that does not have rubber, and the following three designs are with the 3, 5, and 10 percent of rubber, including mixtures. Table 2 shows the final dosages for the four designs.

The concrete mixes were prepared according to the designs. The first mix is the nominal concrete mixture; the others contained 3%, 5%, and 10% recycled rubber.

It is essential to mention that the concrete mixer is Brand CF GUILCO with Model/Series 69,000–59,010, which has a maximum capacity of 30 L. Therefore, the concrete mix design shown in Table 2 is for 30 L. This means that these quantities shown must be multiplied by three to obtain the desired volume so that two beams in 150 × 300 and fourteen 100 × 200 cylinders are cast. Before starting the casting, the first thing to do is to have all the materials weighed. Then, each concrete design is mixed for two minutes, adding aggregates, the cement, three-quarters of the water, the remaining water with additives, and the rubber, respectively.

Once the concrete specimens are prepared after the mixing step, the concrete samples are placed in the individual molds. According to the ASTM C31 [47], each mold has its procedure for molding. For the beams, the concrete must be placed in two layers. The first layer is up to the middle of the mold, then the concrete is compacted 54 times with a rod that has one or two hemispherical tips, 16 mm in diameter and 600 mm in length, then the next layer is placed, and the concrete is compacted again, and finally the mold must be hit with a rubber hammer. The process for cylinder molds is similar; the ones of 100 × 200 are placed in two layers, and compaction is carried out as mentioned above, with the difference that they are only 25 times, and in each layer the mold is hit with a rubber hammer. The cylinders of dimensions of 150 × 300 are the same process, but instead of two layers, there are three.

Once the specimens are processed, after 24 ± 8 h the concrete can be unmolded, coded, and later put into the curing room. ASTM C192 [48] states that the specimens are cured in water tanks until the day they are tested.

### 2.3. Fresh Concrete Tests

The test method for the slump is carried out to determine the mix’s workability. Following the recommendations of ASTM C143 [49], the tools needed for the test are a slump cone, a flat mat absorptive surface, a scoop, a tape measure, and a 5/8-inch diameter tamping rod with hemispherical ends. Through the ASTM C138 [50] standard, two properties of concrete are determined: density and air content.

### 2.4. Hardened Concrete Tests

Once the cylindrical specimens of 100 × 200 reach the desired age, i.e., 7, 14, and 28 days old, the test is carried out to determine the compressive strength. This test is carried out in two specimens for each age so that it can be determined at an average resistance. Following the directions in ASTM C39 [51], where it indicates that the specimen must first be measured twice across the diameter and averaged, then the next step is to put the unbonded caps at the bottom and the top of the specimen and finally center it in the compressive chamber of the testing machine. Once the specimen is ready to be tested, start applying the load until the point of failure. The same testing machine is used to perform the indirect tension testing due to the difficulty of applying a uniaxial tension load on the specimen. According to the ASTM C496 [52], the results obtained from this test may be higher than those from a uniaxial test. The procedure is similar to the compressive test; the difference is that the specimen goes into the testing machine horizontally in a supplementary bearing bar with two strips at the top and the bottom. Finally, the load is applied until the specimen fails.

The beam specimens were subjected to bending tests at 7 and 28 days. The three-point loading tests were carried out according to ASTM C78 [53], which consists of two loading blocks on top that cause a compressive strength and two on the bottom that cause a tensile strength. The first step is to place the specimen in the testing machine, measure, mark the beam, and apply the load continuously until the specimen fails.

Two cylindrical specimens of 100 × 200 mm aged for 28 days are tested to determine the modulus of elasticity according to the ASTM 469 standard [54]. The procedure is to attach the compressometer to the specimen, load it with at least 40% of the compressive strength to set the gauge, and then set the speed.

The cylindrical specimen of 150 × 300 mm is used to prepare the samples aged for 28 days and perform non-destructive tests to determine the behavior and quality of the concrete. The ultrasound instrument is equipment that consists of two transducers connected to the central system: one emits ultrasonic pulses, and the second is the receiver. Considering the recommendations of ASTM C597 [55], the transducers are placed at the ends of the cylinder so that the medium through which the pulse travels is the cementitious matrix, which will prevent or allow the pulse to reach the other end. The result obtained is the time it takes for the wave to arrive from one extreme to the other. This test is used to analyze the uniformity of the concrete; however, correlations can also be made to determine the compressive strength, the modulus of elasticity, and the dynamic modulus of poison. Another non-destructive test performed in the same specimen is the rebound number, a test that allows one to identify weak and strong areas of density in a concrete surface and verify its quality. According to ASTM C805 [56], the test is done through a rebound hammer which acts in such a way that it releases and strikes a steel plunger to be in contact with the surface of the concrete. The tests consist of impacting the surface of the specimen with the hammer ten times, and each impact shall not be closer than 25 mm.

Inspect FEI scanning electron microscopy (SEM) was used for microstructural analysis to observe microstructural patterns in rubber-incorporated concrete. Once samples were tested, fractured pieces were collected and kept under ethanol to prevent hydration and carbonation. After drying for 24 h within a desiccator, the dried samples were put inside the SEM sample chamber to proceed to analysis.

## 3. Results

### 3.1. Fresh Concrete Properties

Figure 3 shows the mix design procedure used in this research. Then the slump, air content and density were obtained (Table 3).

#### 3.1.1. Slump

As seen in the results presented in Table 3, almost all the designs have the same slump, equal to 220 mm, except for the design with 3% of rubber which differs from the others by 2.27%. It is worth mentioning that all the mix designs fulfil the Ecuadorian standards, and it was not necessary to add water. It seems that the designs differ slightly, concluding that the presence of rubber within the mixture does not alter the workability of the concrete. Regarding the results obtained, according to the literature [57], if the slump is between 150 and 220 mm it is considered fluid concrete. These results prove the consistency of the initial design used. It was considered the travel time of the concrete to work that makes the mixture harden until it reaches the design value of 180 mm of the slump to be used in construction. According to the literature [29], the workability of concrete decreases as more rubber is added. However, this refers to rubber as an additive [29], not as a replacement for fine aggregate as in this case.

#### 3.1.2. Air Content

Table 3 shows the air content between 1.3% and 2% in the four designs. According to the designed method [46], the percentage of air content for concrete with a maximum nominal aggregate size being 19 mm is equal to 2%, hence the result of the nominal concrete meets the standard. However, as more rubber is added, the air content decreases. The air content of the design with 3% rubber does not vary with respect to the nominal concrete, and this is because an insignificant amount has been added to alter the air content.

#### 3.1.3. Density

According to the literature [58,59], the more rubber added into the mixture, a block of concrete with a lesser density is obtained. The reason for this phenomenon is attributed to the difference between the specific weight of the aggregate and the rubber. However, there are other researchers who highlight that the density of tires depends on different factors such as manufacturer, location, type, and age [60]. In this research the only design that decreased its density was the 3% design, while the 5% and 10% designs remained the same as nominal.

### 3.2. Hardened Concrete Properties

#### 3.2.1. Compressive Strength

Table 4 and Figure 4 show the average values of the results obtained from the cylindrical specimens 100 × 200 at ages 7, 14, and 28 days. These figures highlight that the compressive strength obtained in all designs is above the design strength, even for nominal concrete, which means that the initial design was oversized. The results obtained are within the same range, which means that replacing a percentage of the fine aggregate with recycled rubber does not significantly alter the compressive strength.

Bisht and Ramana [61] indicated that the presence of rubber in the concrete decreased the compressive strength. However, the difference between nominal concrete and the other three designs is only 5%.

#### 3.2.2. Flexural Strength

The presence of rubber as sand replacement in samples causes a slight increase in flexural strength if the average results are obtained from the beam specimens (see Table 4). The design that obtained the highest flexural strength was with 3% rubber, with a difference of 5.60% with the nominal concrete. The results of the 3% and 5% rubber designs are very similar. However, the design with 5% rubber is far from the two designs mentioned, probably due to some error during the performance of the test; the designs with 3% and 10% recycled rubber only have a difference of 0.37 %, which is similar to the specimen of 5%. The results obtained corroborate what is found in the literature [29], indicating that the rubber increases the resistance to flexion.

#### 3.2.3. Tensile Strength

Figure 5 shows the trend of the results obtained from the tensile strength on the cylindrical specimens of 100 × 200. Contrary to the results obtained from the flexural test, in the tensile strength test the sample with a greater resistance was that of nominal concrete, presenting 5.15 MPa with a 10.45% difference between 3% and 10% of rubber content. In this case, the design with 5% recycled rubber showed the closest result compared to nominal concrete, with a difference of 1.57%. According to the literature [62], the tensile strength in rubberized concrete tends to be lower than in standard concrete because of the weak bond between the cement matrix and the rubber [62].

#### 3.2.4. Modulus of Elasticity

Unlike the results obtained in tensile strength tests these results vary significantly, and it is evident that as more rubber is added the modulus of elasticity decreases (Table 5). There is a decrease of approximately 3.5% and 7.5% between nominal concrete and the designs with rubber contents of 3% and 5%, respectively. However, for the design with 10% of rubber content there is a significant decrease of 30%.

#### 3.2.5. Pulse Velocity

The values in Table 6 indicate that the three designs have similar results for the pulse velocity. However, the one that differs from the rest is the composite with rubber content of 3%, which means that it had the shortest time range from one side to the other side of the specimen. Despite the difference of 14% between the 3% rubber design and the remaining three designs, all four designs fall within the quality category between very good and excellent because the velocity is above 4.5 km/s [63].

#### 3.2.6. Rebound Number

The rebound hammer test was carried out on the 150 × 300 cylindrical specimens for each design at 28 days, and the results are shown in Table 6. The result obtained for the four designs was above 40, which according to the general guideline [63] means the quality of concrete is considered very good. As expected, the compressive strength results for this test are above the design strength. However, these are similar to those obtained in the compression test.

#### 3.2.7. Microstructural Analysis

Microstructural analysis was performed using SEM to identify potential interactions and patterns of rubber particles inside the samples as they were used as a partial replacement for fine aggregate, the sand for 0%, 3%, 5%, and 10%.

The microstructure of selected samples and the distribution of rubber particles and their interface within the cement matrix are shown in Figure 5. In the case of nominal samples, as seen in Figure 6a, a porous aggregate, mainly composed of limestone, is observed on a compact matrix. Figure 6b–d shows the rubber composite concrete with 3%, 5%, and 10% of rubber replacement, respectively. It is worth noting that the transition zone between rubber and cement matrix is seemingly compact among all selected samples. Figure 6 shows a microporous structure of the sample that contains 10% of recycled rubber particles. The pores formed might be attributed to the sample preparation where the concrete vibrating table was not used, and the non-homogeneous distribution of the rubber particles inside the concrete matrix. In addition, all the rubber–concrete composite images show that the shape of rubber particles is quite irregular (as seen in Figure 1). To sum up, SEM analysis shows the dispersion and increasing concentration of recycled rubber with respect to its increasing quantity inside the concrete composite samples. Considering the compressive strength values (see Table 6), rubber particles did not affect the workability and mechanical properties of composite concrete. Hence, recycled rubber particles can be used successfully as replacements for fine aggregate. The SEM image of nominal concrete (Figure 7a) shows a very porous microstructure with a microcrack. On the other hand, SEM images of the concrete composites show the rubber particles inside the concrete matrix and with some cavities except in the sample with 5% recycled rubber. The sample with 5% recycled rubber showed the highest compressive strength among all the samples. As seen in the SEM image of the 5% sample, very small recycled rubber particles have been dispersed inside the matrix, and the microstructure shows two small microcracks. Additionally, no voids or cavities are seen in the microstructure of the sample with 5% recycled rubber, but are in the sample with 10%.

## 4. Life Cycle Assessment (LCA)

Life cycle assessment (LCA) is established in the international normative ISO 14040 [64]. The functional unit analyzed is 1 m^3^ for the concrete mix system. The scope of the assessment evaluates the use of recycled tire rubber for the different mixes. The system has been modeled following scenario A of the ILCD handbook [65]. The scheme begins with the raw material extraction and ends with the concrete mixture produced. Three different mix designs with 3, 5, and 10% recycled rubber were considered and compared to the nominal concrete with no rubber. Data were input into the aiding software, SimaPro 9.0 [66], converting the data to environmental impact indicators. Figure 8 shows the system boundary. The modeling based on the system boundary included the impact of the production of the main component in the concrete mix, which is cement. It also considers the addition of aggregates, both coarse and fine, water, superplasticizer, and the rubber particles. The rubber particles have been modeled as the removal of used synthetic particles from the environment. The used rubber is considered a hazardous waste accordingly, but the modelling reinforces the importance of keeping the rubber particles isolated within the concrete matrix while providing an effect to the mechanical properties of the mix [34].

### 4.1. Inventory

Table 7 shows the inventory for the nominal concrete and the libraries used for the different components for the dosage evaluated. The inventory of the system was based on the dosages given in Table 7. Rubber is set at a value of 0 for the nominal concrete and with a negative sign because of the reuse of discarded tire rubber. The superplasticizer was modeled following the report from EFCA [67].

The method used for this assessment is ReCiPe Method H [68], which allows for a global perspective and analysis of the system that is evaluated. Even though normalization is not defined under ISO 14040 [57], it provides a more straightforward interpretation of the impact analysis results and comparison between scenarios [69]. In addition, the library Concrete 25 MPa, from SimaPro software [66] was used to compare the use of new synthetic and recycled rubbers.

### 4.2. Discussion

The life cycle impact assessment performed for the system and options presented remarkable results in Table 8. From the characterized results, a significant reduction in the carbon footprint can be seen from the nominal concrete to the 10% recycled rubber. This constitutes a 38% reduction in that specific impact indicator. Moreover, less carbon dioxide is emitted with 3 and 5% recycled rubber-containing samples. A review study situated most of the research dealing with used rubber in the low 500 kg CO_2_ eq per m^3^ of concrete mix [70,71]. The values presented in this study show correlation and better performance than the average estimated in the review study. The negative scores in some impact categories correspond to a positive environmental impact which translates into a reduction in the assessed category. This happens because the system considers that the recycled rubber particles are avoiding the toxic component to be released to the ecosystem as they are being encapsulated in the cement binder.

The normalization for the results was also applied in order to get a better under-standing of the environmental performance of the scenarios in the analyzed systems. Figure 9 shows the normalized scores across the 18 impact categories. The most affected categories are ecotoxicity and human toxicity.

However, it is noticeable that the reduction in the impact as the percentage of recycled rubber increases. In fact, for the concrete using 10% recycled rubber, the impacts for the highly affected categories become positive. This behavior is because the reuse of the tires accounts for an end-of-life (EoL) scenario that positively impacts this waste stream. In addition, the dosage has been optimized, which allows for better performance in terms of sustainability. Recycled rubber reuse can replace 3.5 times sand or fine gravel used in concrete systems [72].

The dosage with better environmental performance was the dosage that contemplates recycling 10% of rubber in the mix. A process contribution diagram can exemplify each item in the inventory and its impact on the environment. Figure 10 shows the process contribution for the mix, as mentioned above. The figure shows that recycled rubber positively impacts the 18 impact categories. Furthermore, Portland cement is the major contributor to negative environmental performance. In the water consumption category, sand extraction and cement manufacture have a considerable contribution.

Additionally, the 10% mix was compared to a library found in the SimaPro software [66]. The inventory is for concrete that has a 25 MPa strength and includes the use of synthetic rubber. The score of the library was chosen as the basis of comparison. Figure 11 shows the comparison of the impact categories affected by concrete and concrete with 10% of recycled rubber. As seen, recycled rubber impacts both global warming potential and land use categories more compared with nominal concrete. The impact is only ~13% higher, but the positive impacts in the other categories show an overall better environmental evaluation. The use of waste material in a concrete mix results in environmental credit for the specific system, and can be discussed as an end-of-life scenario for the disposal of hazardous materials. One study concluded that replacing 10% rubber for road pavement was optimal when analyzing different aggregates [72].

## 5. Conclusions

The mechanical properties and environmental impact of recycled rubber as fine aggregate incorporated in fresh, 7-, 14-, and 28-days aged concrete have been investigated. All the specimens were analyzed using destructive tests including compression, tensile, flexural, and modulus of elasticity. In addition, some specimens were used to perform non-destructive tests such as the ultrasound pulse velocity and the rebound number. In addition, a life cycle assessment was made in order to quantify the environmental impact of the mix designs.
Recycled rubber does not significantly alter the mechanical properties of fresh concrete and aged concrete for up to 28 days. Additionally, the use of recycled rubber can be used to reduce the environmental impact;Up to 5% of recycled rubber as a replacement for fine aggregate is feasible considering the mechanical properties of the desired concrete;The life cycle assessment showed a reduction of up to 40% in the global warming potential. In fact, the 15% recycled rubber concrete mix has a climate change indicator of approximately 245 kg of CO_2_ eq, which means that the global warming potential significantly decreases with the increase in the amount of recycled rubber used in the concrete as fine aggregate.

## Figures and Tables

**Figure 1 materials-16-00063-f001:**
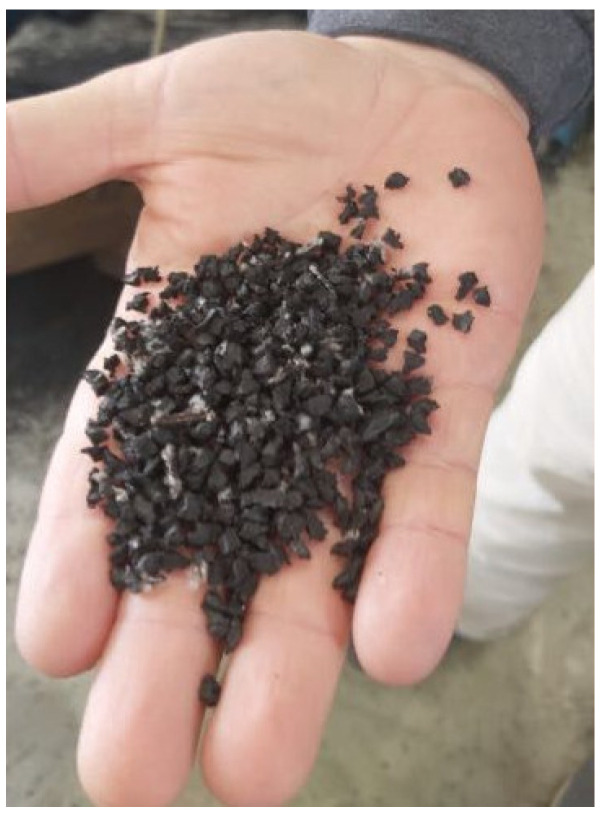
Ground rubber aggregate.

**Figure 2 materials-16-00063-f002:**
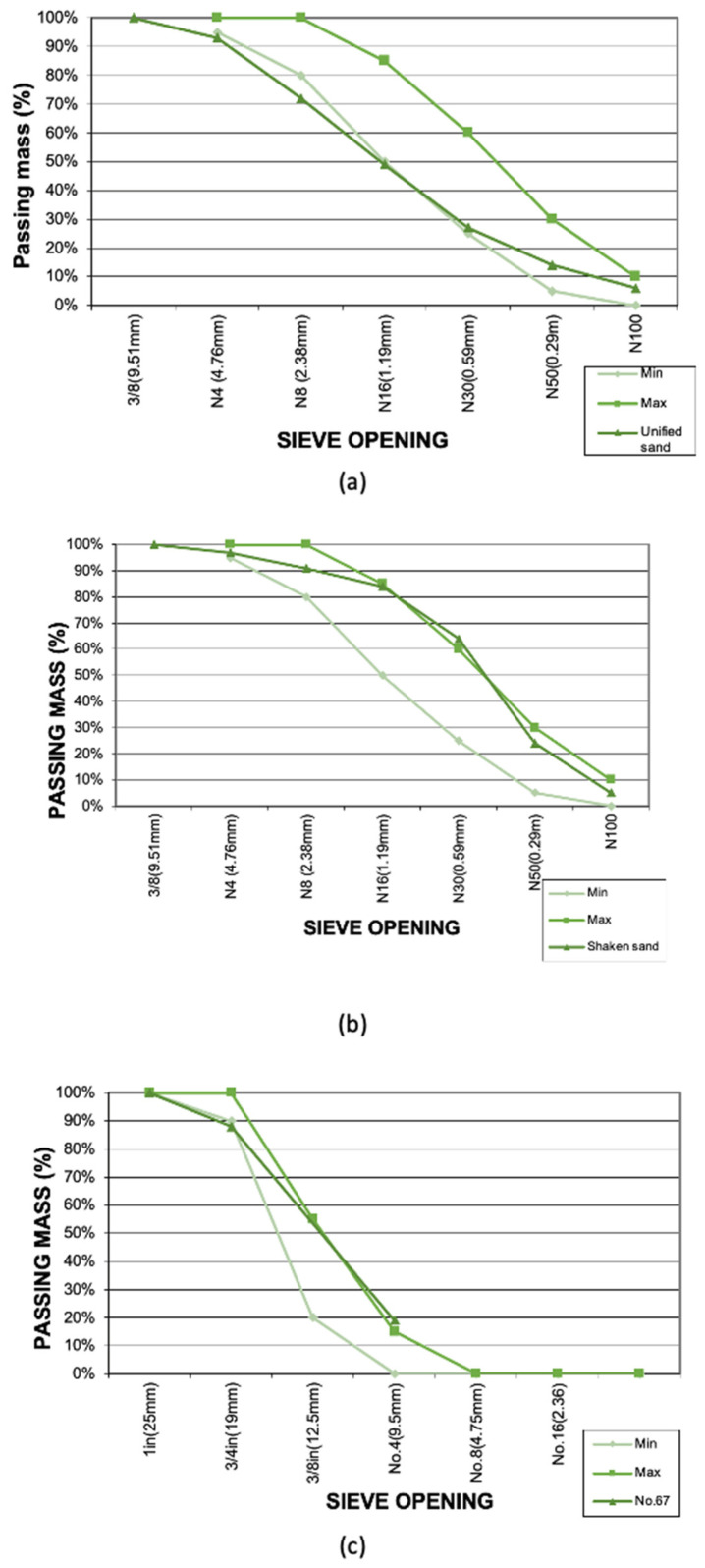
The granulometric curve of: (**a**) river sand; (**b**) crushed sand; (**c**) aggregate No.67.

**Figure 3 materials-16-00063-f003:**
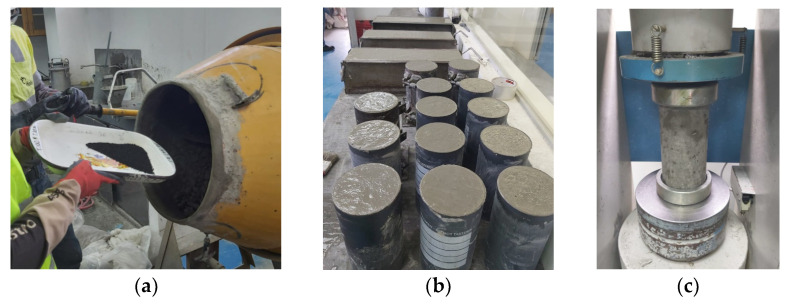
(**a**) rubber added in concrete mix design; (**b**) Concrete samples; (**c**) Compressive strength test.

**Figure 4 materials-16-00063-f004:**
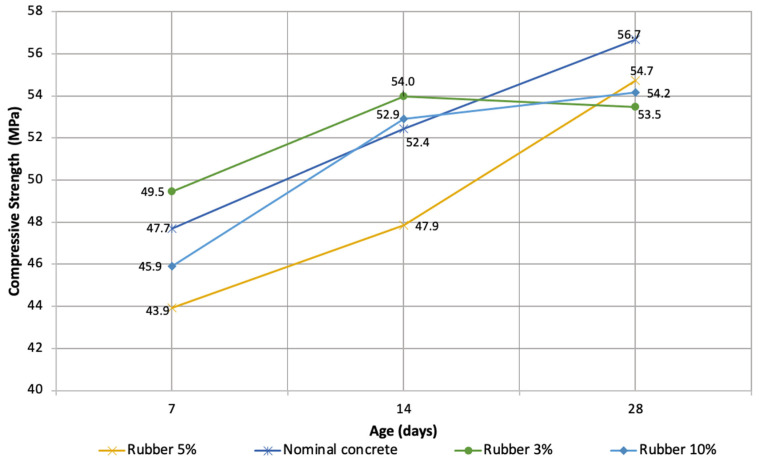
Compressive strength at the ages of 7, 14, and 28 days.

**Figure 5 materials-16-00063-f005:**
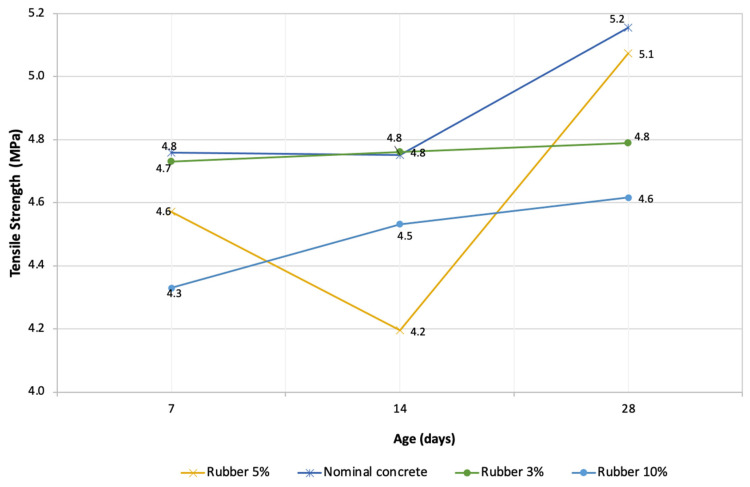
Tensile strengths at the ages of 7, 14, and 28 days.

**Figure 6 materials-16-00063-f006:**
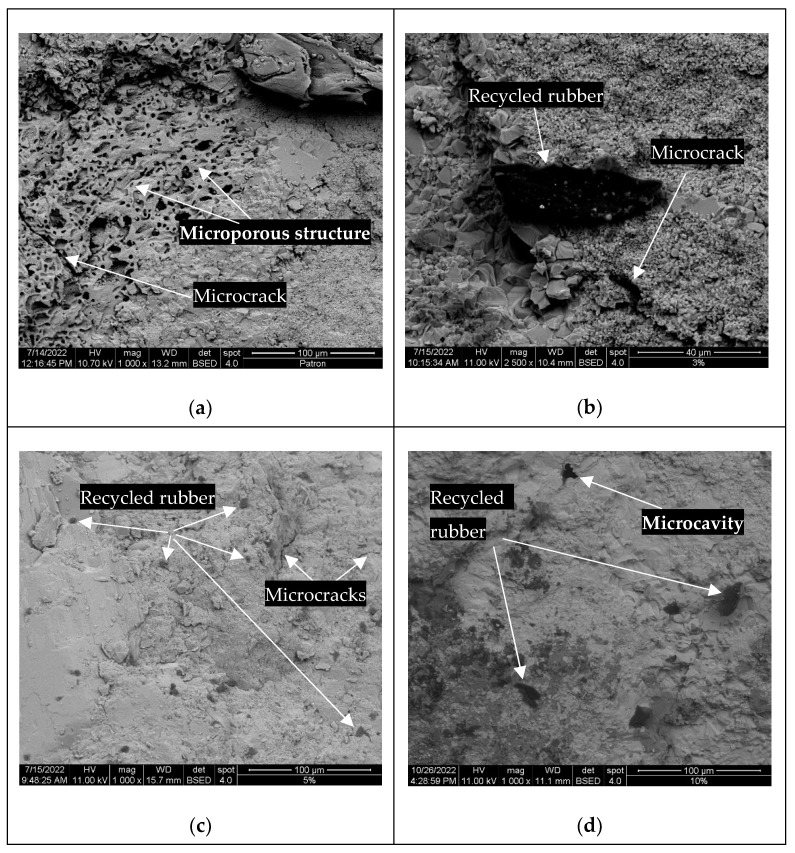
SEM images of concrete mixed with ground rubber particle in different proportions: (**a**) 0%; (**b**) 3%; (**c**) 5%; and (**d**) 10%.

**Figure 7 materials-16-00063-f007:**
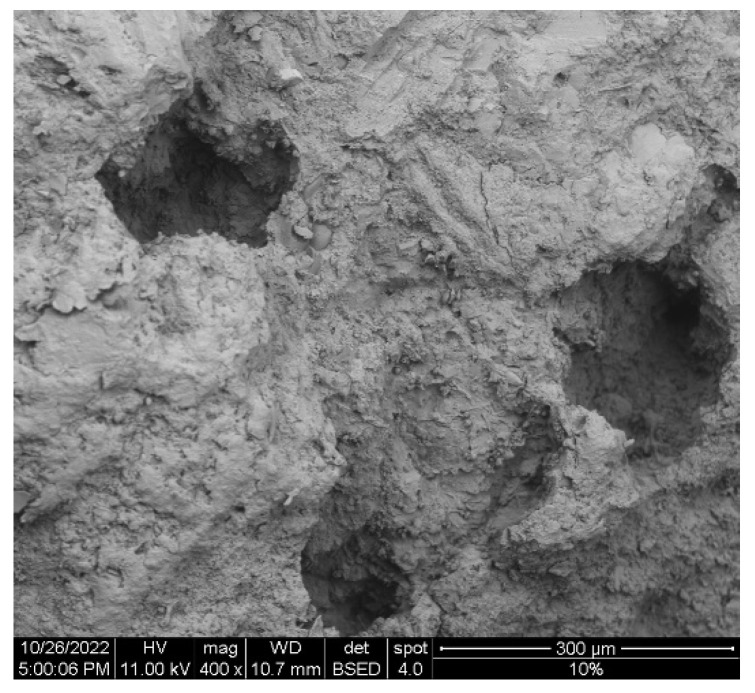
Microporous structure of the sample reinforced with 10% recycled rubber.

**Figure 8 materials-16-00063-f008:**
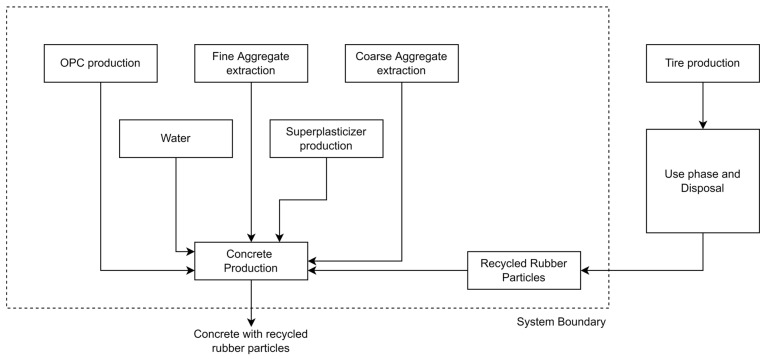
System boundary for mixes using recycled rubber.

**Figure 9 materials-16-00063-f009:**
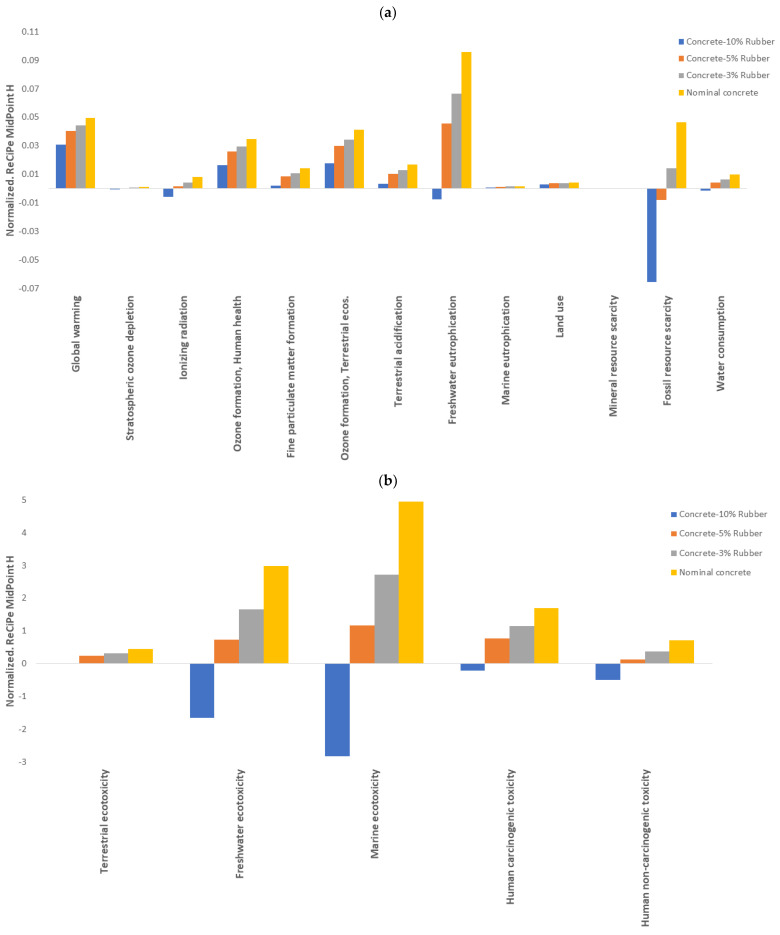
Normalized results for 1 m^3^ of the concrete mixes: (**a**) low impact categories; and (**b**) higher impact categories.

**Figure 10 materials-16-00063-f010:**
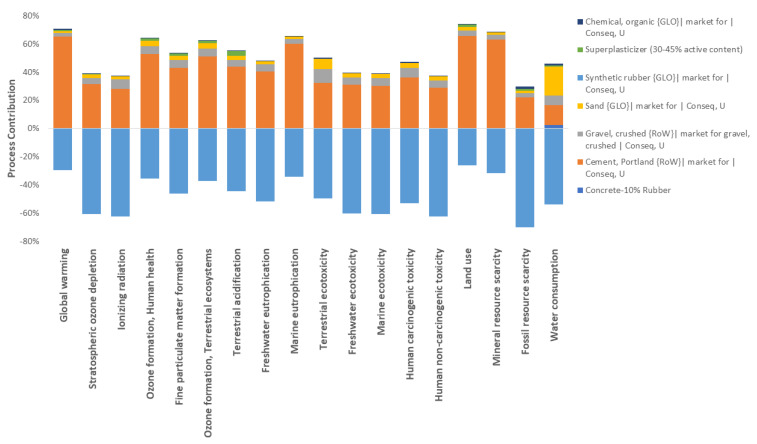
Process contribution for the 10% recycled rubber dosage.

**Figure 11 materials-16-00063-f011:**
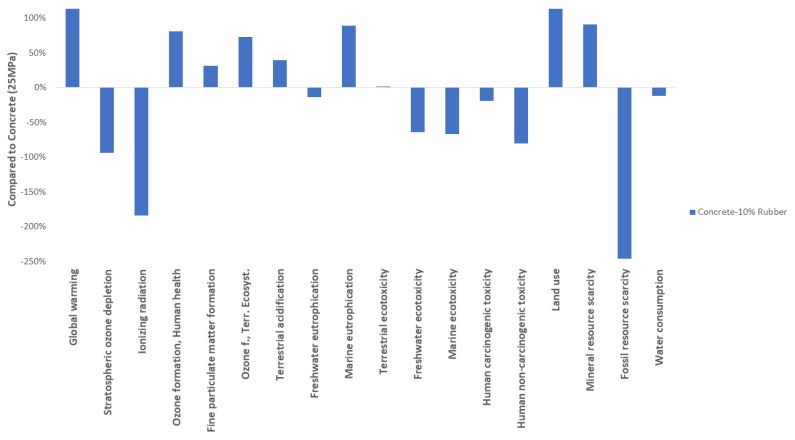
Percentual environmental impact of concrete that contains 10% of recycled rubber compared to 25 MPa Concrete.

**Table 1 materials-16-00063-t001:** Chemical composition and physical properties of the cement HE given by Holcim.

Chemical Composition	Portland cement clinker Gypsum Natural pozzolans
Physical properties	Change in length per autoclave −0.04% Setting time, Vicat method: 150 min
	Air content of the mortar: 3% Minimum compressive strength 1 day: 14 MPa 3 days: 25 MPa 7 days: 32 MPa 28 days: 40 MPa

**Table 2 materials-16-00063-t002:** Concrete mix design (kg for a 28 lt. batch).

	Nominal Concrete	Rubber 3%	Rubber 5%	Rubber 10%
Cement	10.92	11.06	11.85	11.85
No.67	24.63	24.94	27.02	26.75
River sand	17.79	16.73	17.06	14.88
Crushed sand	8.65	8.30	8.38	7.27
Rubber	0.00	0.53	0.95	1.89
Water	3.92	3.65	3.75	4.13
Superplasticizer	0.055	0.055	0.059	0.059
Retarder	0.131	0.133	0.142	0.142

**Table 3 materials-16-00063-t003:** Results of fresh concrete tests.

Property	Slump [mm]	Air Content [%]	Density [kg/m^3^]
Nominal	220	2.0	2360
3%	215	2.0	2355
5%	220	1.5	2360
10%	220	1.3	2364

**Table 4 materials-16-00063-t004:** Mechanical properties through destructive testing.

Property	Compressive Strength (MPa)			Splitting Tensile Strength (MPa)			Flexural Strength (MPa)	
Age (days)	7	14	28	7	14	28	7	28
Nominal	47.679	52.422	56.679	4.759	4.752	5.155	4.46	5.05
3%	49.456	53.957	53.481	4.731	4.761	4.789	4.21	5.35
5%	43.939	47.863	54.720	4.571	4.195	5.074	4.78	4.69
10%	45.892	52.910	54.159	4.330	4.532	4.616	5.4	5.33

**Table 5 materials-16-00063-t005:** Results of the modulus of elasticity test.

Design	Modulus of Elasticity [GPa]
Nominal concrete	32.91
3%	31.76
5%	30.43
10%	23.15

**Table 6 materials-16-00063-t006:** Non-destructive testing results.

Property	Ultrasonic Pulse Velocity (km/s)	Rebound Number	Average Compressive Strength (MPa)
Nominal	4.504	50.00	56.50
3%	5.239	49.80	56.20
5%	4.518	54.00	63.20
10%	4.500	50.80	57.80

**Table 7 materials-16-00063-t007:** Inventory for the nominal concrete.

**Products**		
Nominal concrete	66.096	kg
**Resources**		
Water, unspecified natural origin, EC	3.92	l
**Materials/fuels**		
Cement, Portland {RoW}|market for|Conseq, U	10.96	kg
Gravel, crushed {RoW}|market for gravel, crushed|Conseq, U	24.63	kg
Sand {GLO}| market for|Conseq, U	26.44	kg
Synthetic rubber {GLO}|market for|Conseq, U	0	kg
Superplasticizer (30–45% active content)	0.131	kg
Chemical, organic {GLO}|market for|Conseq, U	0.055	kg

**Table 8 materials-16-00063-t008:** LCA results for recycling rubber from tires used in 1 m^3^ of concrete.

Impact Indicator	Unit	Concrete-10% Rubber	Concrete-5% Rubber	Concrete-3% Rubber	Nominal Concrete
Global warming	kg CO_2_ eq	2.44 × 10^2^	3.34 × 10^2^	3.46 × 10^2^	3.96 × 10^2^
Stratospheric ozone depletion	kg CFC11 eq	−1.97 × 10^−5^	2.14 × 10^−5^	3.55 × 10^−5^	7.59 × 10^−5^
Ionizing radiation	kBq Co-60 eq	−5.45	−1.52	5.78 × 10^−2^	3.96
Ozone formation, Human health	kg NO_x_ eq	4.95 × 10^−1^	7.22 × 10^−1^	7.59 × 10^−1^	7.17 × 10^−1^
Fine particulate matter formation	kg PM2.5 eq	8.84 × 10^−2^	2.50 × 10^−1^	2.94 × 10^−1^	3.64 × 10^−1^
Ozone formation, Terrestrial ecosystems	kg NO_x_ eq	4.79 × 10^−1^	7.28 × 10^−1^	7.73 × 10^−1^	7.31 × 10^−1^
Terrestrial acidification	kg SO_2_ eq	2.10 × 10^−1^	4.84 × 10^−1^	5.55 × 10^−1^	6.92 × 10^−1^
Freshwater eutrophication	kg P eq	1.74 × 10^−2^	4.76 × 10^−2^	5.58 × 10^−2^	6.23 × 10^−2^
Marine eutrophication	kg N eq	−1.69 × 10^−3^	5.54 × 10^−4^	1.36 × 10^−3^	7.78 × 10^−3^
Terrestrial ecotoxicity	kg 1,4-DCB	−9.41	2.11 × 10^2^	2.79 × 10^2^	4.66 × 10^2^
Freshwater ecotoxicity	kg 1,4-DCB	−4.68	−2.44 × 10^−1^	1.44	3.66
Marine ecotoxicity	kg 1,4-DCB	−6.22	−2.39 × 10^−1^	2.03	5.11
Human carcinogenic toxicity	kg 1,4-DCB	−1.82	1.45	2.58	4.73
Human non-carcinogenic toxicity	kg 1,4-DCB	−1.24 × 10^2^	3.79	5.17 × 10^1^	1.06 × 10^2^
Land use	m^2^a crop eq	6.96	1.20 × 10^1^	1.31 × 10^1^	2.66 × 10^1^
Mineral resource scarcity	kg Cu eq	8.30 × 10^−1^	1.15	1.19	2.05
Fossil resource scarcity	kg oil eq	−6.62 × 10^1^	−6.88	1.60 × 10^1^	4.57 × 10^1^
Water consumption	m^3^	−1.37	2.63 × 10^−1^	8.71 × 10^−1^	2.64

## Data Availability

Not applicable.

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
