# Peer review of "Effects of Recycled Rubber Particles Incorporated as Partial Sand Replacement on Fresh and Hardened Properties of Cement-Based Concrete: Mechanical, Microstructural and Life Cycle Analyses"

_materials, 2022, doi:10.3390/ma16010063_

Round 1

Reviewer 1 Report

The selected topic has certain significance in the recycling of waste, but there are still some problems about the format and content that need further revision and consideration.

1. Generally, the abstract should include the research background.

2. Please provide more informtion about the rubber aggregate, such as its particle size gradation.

3. As shown in Table 3, the slump of concrete with 3% rubber content is different from others. The result is not explained. And Table 3 is misplaced.

4. The article does not include the process of producing recycled rubber into the system boundary to consider the environmental impact, which is illogical.

5. The reasons for the negative values in Table 8 are not explained clearly.

6. The performance of concrete with 10% rubber content is weaker than that of concrete with 5% rubber content, but it has less impact on the environment. There should be a value that balances them in the range of 5%-10%, so 3 %, 5%, 10% gradients are inadequate.

7. The conclusions should be supported by some quantitative results.

8. Figure 3 has no title.

9. It’s difficult to read the words in Figure 5, they are not well displayed. Some other Figures are also needed to be improved, such as Figures 8-10.

Author Response

The authors are very grateful to the reviewers for the revision of this manuscript, their assessments, constructive comments, and the time they have spent on reviewing our manuscript. The comments from the reviewer (black bold font) and our responses (blue font) are included below. The modified actions are in red font in the revised manuscript.

In addition to the corrections requested by the reviewers, the authors have undertaken substantial revision of the manuscript and hope that this revised manuscript can be accepted for publication in Materials Journal MDPI.

Reviewer 2 Report

The results of “Effects of recycled rubber particles incorporated as partial sand replacement on fresh and hardened properties of cement-based concrete: Mechanical, microstructural and life cycle analyses” are of potential interest. The introduction section provides sufficient background of past literatures. In the experimental Programme section, all the testing methods are sufficiently described. In the experimental result and discussion section, the results are elaborately discussed with figures and tables. The conclusions are well presented and it is supported by the results. All the references are related to this research and also sufficient. However, the following major corrections are to be carried before the acceptance of the Manuscript.

1. Abstract: State the need of the study. Merge 2 paragraphs into one. Present the your research recommendation.

2. Key words: Remove the numerals 1, 2, 3, 4

3. Cite the sentence ‘The raw materials for cement as well as aggregates that come from natural resources are generally over-exploited [5]’. Some works are found below.

https://doi.org/10.3390/ma15124272

https://doi.org/10.1016/j.conbuildmat.2010.12.026

https://doi.org/10.12989/sem.2022.83.3.387

4. Cite the sentence ‘Specifically, Portland cement is associated with high carbon dioxide emissions, responsible for 3.8% of the net global greenhouse gas, representing c.a. 5% of global carbon dioxide emissions [6]’. Some work are as follows.

https://doi.org/10.1504/IJESD.2019.099491

https://doi.org/10.17533/udea.redin.20190403

https://doi.org/10.1002/suco.201900162

https://doi.org/10.1016/B978-0-12-821730-6.00031-0

5. What is the novelty of your research?

6. Show the experimental photos in the manuscript.

7. Figure caption is available for compressive strength chart. Also legend is not available. It is better to prepare a bar chart for various mixtures, indicating 7,14,28 days strength.

Author Response

(The authors gave the same response as above.)

Round 2

Reviewer 1 Report

The manuscript has been well revised, and it is recommended to be accepted for publication.

Reviewer 2 Report

The author addressed all the comments raised by me. It can be published in the present form.